# An active machine learning framework for automatic boxing punch recognition and classification using upper limb kinematics

Saravanan Manoharan[1], John Warburton[2], Ravi Sadananda Hegde[3], Ranganathan Srinivasan[4], Babji Srinivasan[1]*

1 Department of Applied Mechanics and Biomedical Engineering, Indian Institute of Technology Madras, Chennai, India, 2 Applied Sport and Exercise Science, Liverpool John Moores University, Liverpool, United Kingdom, 3 Department of Electrical Engineering, Indian Institute of Technology Gandhinagar, Gandhinagar, Gujarat, India, 4 Department of Chemical Engineering, Indian Institute of Technology Madras, Chennai, India

* babji.srinivasan@iitm.ac.in

## Abstract

Boxing punch type classification and kinematic analysis are essential for coaches and athletes, providing critical insights into punch variety and effectiveness, which are vital for performance improvement. Existing methods for punch recognition and classification typically rely on wearable sensor data or video data; however, no fully automated system currently exists. While coaches prefer video-based analysis for its ability to easily visualize punch action errors and refine technique, video-based classification suffers from lower accuracy compared to sensor-based methods due to limitations such as motion blur. Current classification approaches typically employ supervised learning, requiring experts to annotate 70–80% of the data for model training. However, the high sampling frequency of sensor data makes this process time-consuming and challenging, leading to potential fatigue and an increased risk of inconsistent annotations by domain experts. This paper proposes a novel multimodal approach that integrates wearable sensor data and video data for automatic punch recognition and classification. The method also includes automatic segmentation of punch videos, which improves classification accuracy by utilizing both data sources. To reduce labeling effort, we apply a Query by Committee-based active learning technique, significantly decreasing the required labeling effort by one-sixth. Using only 15% of the typical labeling effort, our system achieves 91.41% accuracy for rear-hand punch recognition, 91.91% for lead-hand punch recognition, and 92.33% and 94.56% for punch classification, respectively. This Smart Boxer system aims to enhance punch analytics in boxing, providing valuable insights to improve training, optimize performance, and increase fan engagement with the sport.

**Data availability statement:** The data supporting the findings of this study are available in the Zenodo repository at https://doi.org/10.5281/zenodo.14965635. This dataset includes IMU sensor data for various boxing punches, such as Lead Hand Jab, Lead Hand Hook, Lead Hand Uppercut, Rear Hand Jab, Rear Hand Hook, and Rear Hand Uppercut, with a sampling frequency of 200 Hz.

**Funding:** This research was supported by the Centre of Excellence for Sports Science and Analytics for funding from the Indian Institute of Technology, Madras, under Grant SP22231231CPETWOSSAHOC. The funders had no role in study design, data collection and analysis, decision to publish, or preparation of the manuscript.

**Competing interests:** The authors have no relevant financial or non-financial interests to disclose.

**Abbreviations:** IMU: Inertial Measurement Unit; KPI: Key Performance Indicator; IoT: Internet of Things; HAR: Human Action Recognition; AI: Artificial Intelligence; QBC: Query By Committee; KNN: K- Nearest Neighbor

# 1. Introduction

In recent years, the sport of boxing has undergone a transformative evolution driven by technological innovations such as computer vision and IoT wearables. These advancements have revolutionized how boxers train, compete, and analyze their performance, marking a significant shift towards data-driven sports approaches. By integrating computer vision techniques, boxing training regimens have become more precise and personalized, allowing coaches and athletes to extract invaluable insights from video analysis. Furthermore, incorporating IoT wearables has facilitated real-time monitoring of vital physiological parameters, enabling trainers to optimize training protocols and minimize injury risks.

## 1.1. Computer vision based punch analytics system

Human Action Recognition (HAR) in sports is critical for detecting and identifying player actions in training and competitions. In sports, actions vary in complexity and interaction levels, necessitating a systematic approach to classification. Challenges remain, such as the fixed number of input frames, which may limit detection accuracy for actions requiring extended temporal context [1].

In boxing, HAR is employed to analyze key performance indicators (KPIs), including the type and frequency of punches (jab, hook, uppercut) thrown by the lead and rear hands. This data supports coaching and performance enhancement. A non-invasive method using vision cameras and time-of-flight sensors has been proposed for pose and behavior feature extraction to identify punch types [2]. However, vision cameras may struggle with perpendicular punches or occlusion. To address this, overhead depth imagery has been used, along with robust body-part tracking, to mitigate occlusion-related challenges and improve sensor reliability [3].

## 1.2. Wearable sensor-based punch analytics system

Wearable IoT sensors have become reliable tools for athlete performance analysis across various sports, facilitated by advancements in microelectromechanical systems (MEMS). In combat sports like boxing, Wearable sensors provide objective metrics for key performance indicators (KPIs) such as punch accuracy, power, speed, defense effectiveness, and ring generalship [4–7]. These metrics help coaches and analysts assess strengths, weaknesses, and areas for improvement. IoT sensors, particularly inertial measurement units (IMUs), enable detailed punch analysis by identifying punch types, quantifying total punch count, and capturing the start and end times of punches. This data provides insights into punch speed, efficiency, and technique refinement. Categorizing punches (e.g., jab, hook, uppercut) further aids in performance assessment and strategic planning, supporting targeted skill enhancement [8–13].

This research employs a multi-modal system combining IoT wearables and vision systems to analyze boxing punch performance. IoT wearables deliver accurate quantitative data but lack real-time bout analysis, insufficient visual feedback and qualitative insights [14,15]. Vision systems provide real-time bout analysis but have

motion blur, reduced classification accuracy, multi-camera synchronization, blind spot issues, and lower accuracy, potentially affecting punch classification, as shown in Table 1.

To address these limitations, IoT wearables are used to train a model for automated punch recognition and classification, as illustrated in Fig 1. The classified data is utilized to segment punch frames from synchronized videos with high accuracy for detailed analysis.

A query-by-committee-based active learning technique minimizes labeled data requirements, using only 15% for training while improving accuracy and adaptability. Unlike deep learning models, which typically require 70–80% of the dataset for training, our approach leverages active learning to minimize data labeling efforts while maintaining high classification accuracy. Deep learning models, such as CNNs and MLPs, often face overfitting issues when training data is limited, whereas our method iteratively refines predictions, improving adaptability. Furthermore, deep learning approaches can be computationally expensive, making real-time processing and post-activity punch analysis challenging without high-end hardware. Our approach is more computationally efficient, ensuring practical implementation for both training and competition scenarios. Also identifies key performance indicators (KPIs), including total punch count, punch start and end times, and punch type categorization, offering detailed insights into a boxer's performance and activity levels.

**Table 1. Literature review.**

| Reference | Sensor & Field view | Algorithm developed | Metrics | Accuracy | Limitation |
|---|---|---|---|---|---|
| **In Computer Vision Technology** | | | | | |
| [16] | Swissranger SR4000 time-of-flight (ToF) camera & 8 | Multi-class SVM and Random forest classifiers | Classify straight, hook and uppercut punches | 96.2% | Sensitivity to lighting conditions |
| [17] | Qualisys & 6, Kistler force plate | Visual 3D software | 3D Kinetics & Kinematics of punch | – | Blind spots or occlusions |
| [4] | High-resolution camera | YOLOv5 model | Scoring accuracy & winner prediction | 63.33% and 70% | Accuracy is low |
| [18] | GoPro Hero8 cameras & 4 | Euclidean measure | Contact between the boxers | – | Real-time processing |
| **In IoT Wearable System Technology** | | | | | |
| [12] | 2 IMU sensors, ATMega328P | KNN, Random Forest & SVM | punch classification | 37.37%-59.16% | Training and labeling more data poses challenges |
| [19] | Smartphone-based Phyphox IMU app | 12-layer deep neural network with TFLearn | Punch classification | 79.2% | Limited data cause overfitting, hard to predict unknown user behavior |
| [13] | IMU sensor | Multilayer perceptron (MLP-NN) | Punch recognition | 91.89% (amateur) & 92.93% (elite) | Applied to practice sessions rather than real-time matches |
| [8] | SABELSense 9DOF IMU | LR,SVM,MLP-NN,RF,XGB | Classify strike type | MLP-NN(98%) | Discomfort During training |
| [20] | Arduino NodeMCU, Vibration, force sensor, and IMU | STATISTICA software™ Statistical analysis | Punching and kicking forces, Punching location with reaction times | – | Applied to practice sessions rather than real-time matches. |
| [21] | Piezoresistive sensors, IMU, Kistler force plate and Vicon Mocap | IBM SPSS Statistical analysis | Punch biomechanics | – | Limited movement of boxer, Isolated Testing environment |

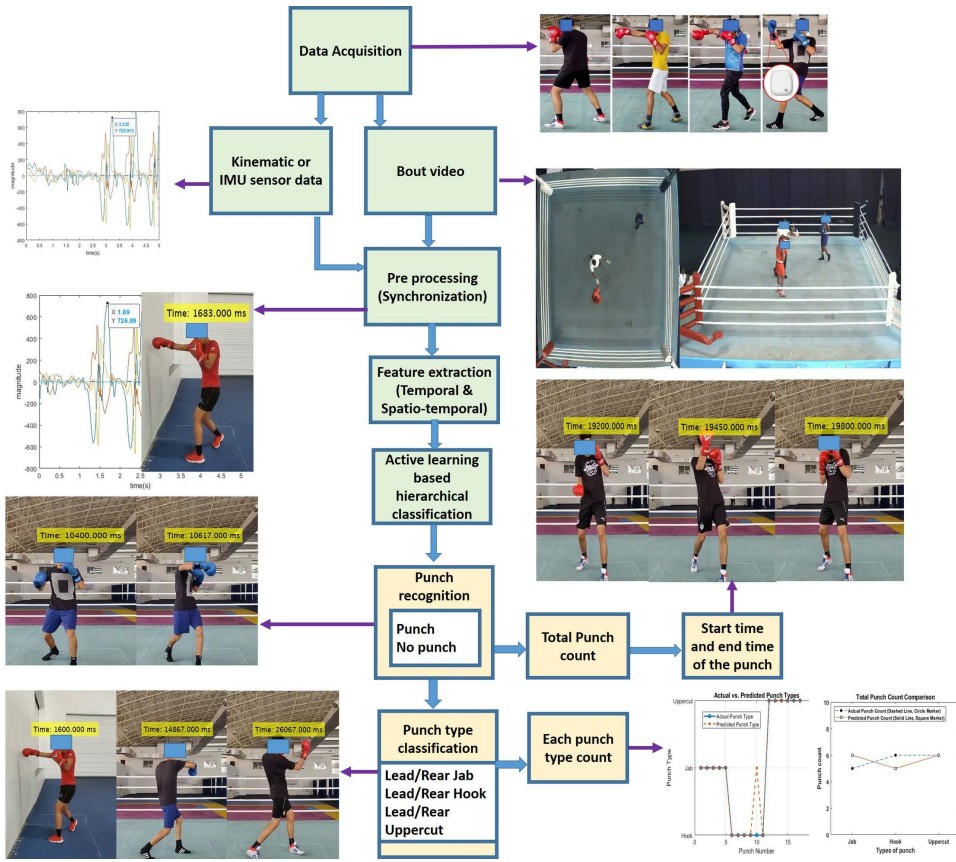

**Fig 1. Flow chart of proposed system.**

## 2. Materials and Methods

At the Inspire Institute of Sports (IIS) in Karnataka, India, data were collected from eight elite boxers, focusing on five orthodox boxers (three males, two females) aged 20–25 years, weighing 57–66 kg, and with heights of 165–175 cm, classified as open-skill class boxers. All participants were adults, and written informed consent was obtained before data collection. The recruitment period for this study was from 1st July 2024–26th July 2024. The study was conducted in accordance with the ethical guidelines issued by the Indian Council of Medical Research, Schedule Y of the Indian Drugs & Cosmetics Act 1940, and Good Clinical Practices Guidelines, as approved by the Inspire Institute of Sport Research Ethics Committee (Ethics number: EC/IIS/2024/020). Using MetaMotion IMU sensors by Mbientlab (±16 g accelerometer, ±2000 deg/s gyroscope, 200 Hz sampling rate), motion data were captured in real time as shown in Fig 2. Sensors were secured on both wrists under gloves and transmitted data via Bluetooth for analysis of punch movements, supporting applications in sports, wearables, and robotics.

### 2.1. Data acquisition

The study involved the simultaneous collection of IMU data (as shown in Fig 3) at 200 sampling frequency and video recordings at 60 FPS for labeling and ground truth validation. Eight elite boxers (five orthodox and three southpaw) performed 320 shadowboxing punches, covering 14 punch types. For analysis, six distinct punch types, including a "no punch" category, were classified. The lead hand punches analyzed included the long-range lead jab to the head (jab),

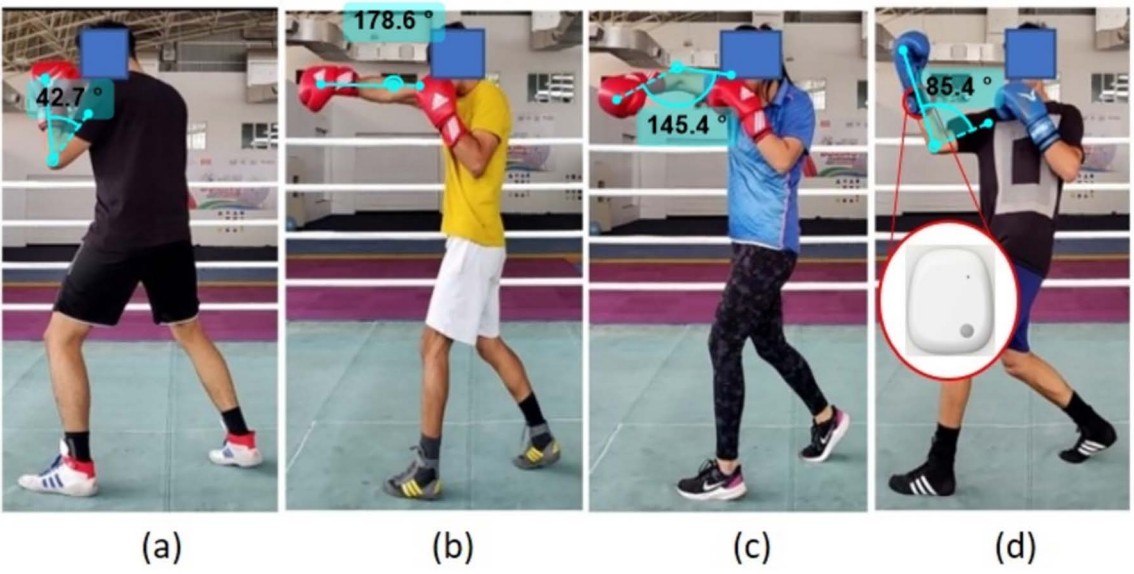

**Fig 2. Punch types and its kinematics:** (a) No punch (b) Long-range rear jab to head (c) Mid-range rear hook to head (d) Mid-range rear upper-cut to head.

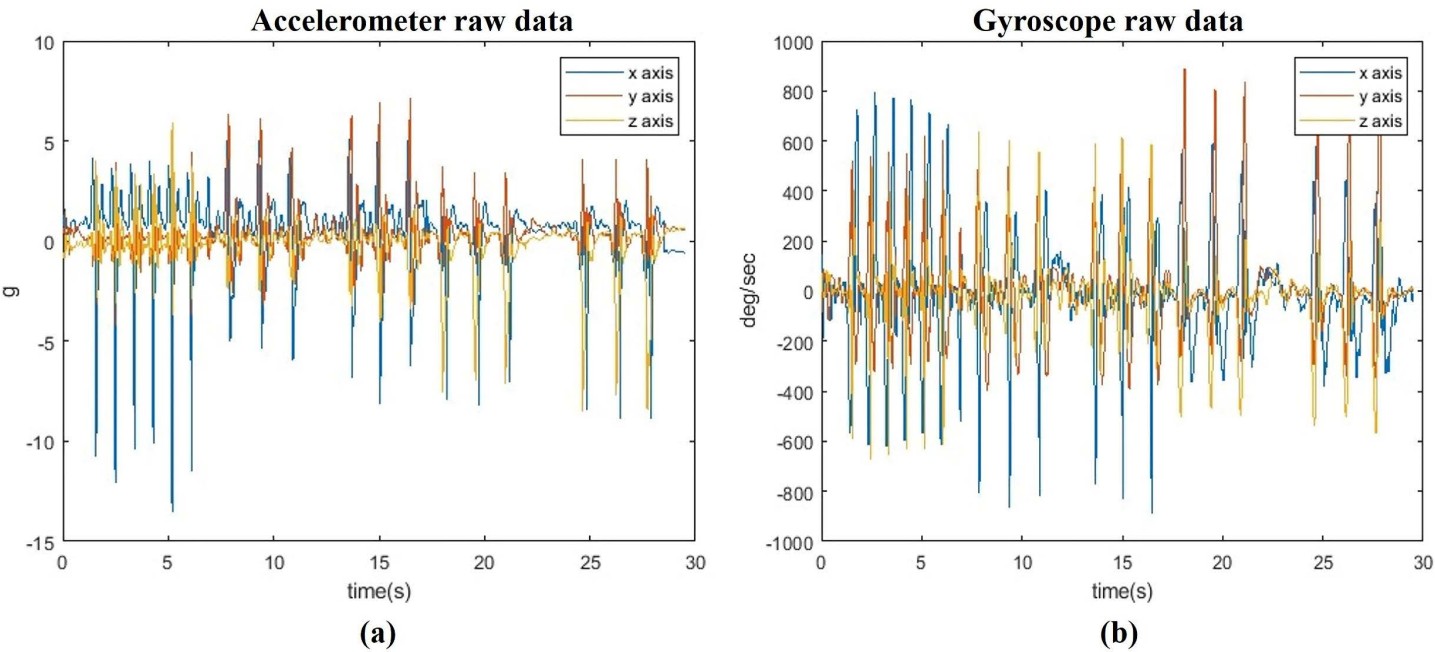

**Fig 3. IMU raw data:** (a) 3-DoF accelerometer data (b) 3-DoF gyroscope data.

mid-range lead hook to the head (hook), and mid-range lead uppercut to the head (uppercut). The rear hand punches examined included the long-range rear jab to the head (jab), mid-range rear hook to the head (hook), and mid-range rear uppercut to the head (uppercut). Video data were captured using a mobile camera synchronized with IMU recordings.

Each boxer performed three punch types (jab, hook, uppercut) in four directions, repeating each punch 40 times (10 punches per side). This resulted in three videos per boxer, totaling 15 videos for the five orthodox boxers. Additionally, three mixed-punch videos from other boxers were recorded for testing purposes, ensuring comprehensive evaluation.

## 2.2. Data pre-processing

After collecting data from the IMU sensor and cameras, the video frames were not synchronized with each other. The video data was captured at a frame rate of 60 Hz, while the IMU data was recorded at 200 Hz. The starting and stopping times of both devices differ. However, both datasets include UTC timestamps, allowing us to address the timing discrepancies. For labeling and ground truth, the Data Capture Lab application was utilized. This application visualizes the data and video according to their respective sampling frequencies, facilitating accurate labeling and ground truth. Fig 4 illustrates a synchronized duration of 1.69 seconds in the plot (IMU data), while the corresponding video frame shows a time of 1683 milliseconds.

## 2.3. Feature extraction

Punch kinematic data were analyzed using accelerometer and gyroscope measurements along the x, y, and z axes, with all punch types completing within an average duration of 0.9 seconds (~180 samples). A rolling window of 180 samples with a 179-sample overlap at a 200 Hz sampling frequency was employed to capture time-frequency characteristics and ensure temporal alignment in the analysis. Feature extraction focused on spatiotemporal characteristics, particularly statistical and time-frequency domain features, to enhance classification accuracy. The selected statistical and time-frequency features were chosen based on the distinct kinematic characteristics of each punch type, ensuring accurate classification: Time-domain features (mean, standard deviation, max, min, interquartile range, entropy, skewness, kurtosis, mean absolute deviation) capture variations in punch intensity, curvature, and rotational control. Frequency-domain features (power spectral density, spectrogram) identify dominant motion patterns across axes, with jabs showing higher spectral density on the x-axis, hooks on the y-axis, and uppercuts on the z-axis. This approach ensures that the extracted features effectively distinguish punch types while preserving key biomechanical patterns.

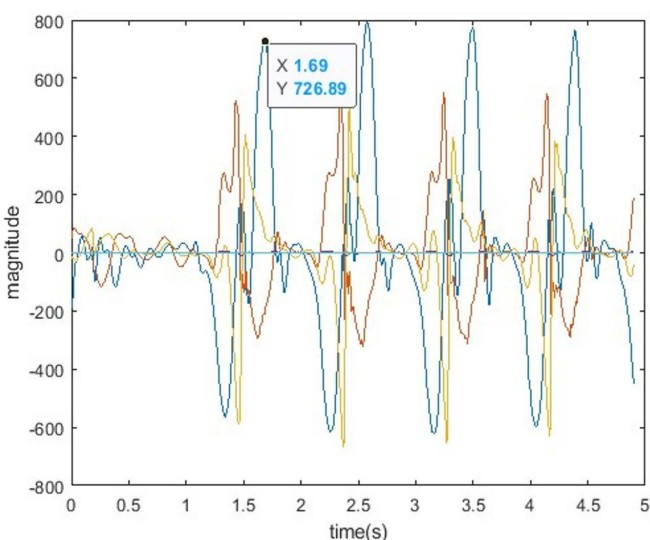 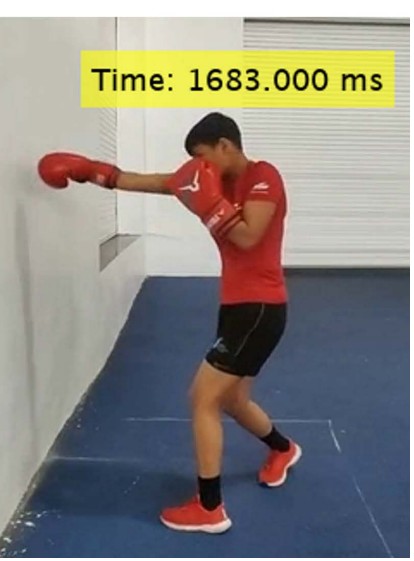

**Fig 4. IMU data and video synchronization.**

## 2.4. Hierarchical classification

In the initial phase of our hierarchical classification system, a binary recognition approach was implemented to detect the presence of punches and determine their start and end times. Punches were further classified into types (jabs, hooks, and uppercuts) using the random forest technique, achieving a classification accuracy of 96%, as demonstrated in previous research [6]. The training dataset included 80% of the data (120 punches per category), while 20% (50 punches per category from unknown boxers) was used for testing. Manual labeling of high-frequency data (200 Hz) by domain experts is resource-intensive and challenging, a common issue in IoT sensor-based classification [5,8,19,20]. To address these challenges, we adopted an innovative approach known as active learning modeling from the literature [22–24]. Unlike traditional machine learning methods that often require large portions of the dataset, typically 70% to 80%, for training, active learning significantly streamlines this requirement. It accomplishes accurate classification with just a fraction of the data, such as 18 punches from each category like jab, hook, and uppercut, around 15%. This approach reduces the computational burden and minimizes the cost and time of labeling an extensive dataset.

Data labeling was conducted using video data as the ground truth, employing a percentage-based criterion ranging from 10% to 100%. Specifically, under the 60% criteria, instances were labeled as "punch" if 60% of the punch occurred within the fixed-length window of 0.8 seconds (average punch action time) or 160 samples; otherwise, they were labeled as "no punch." This labeling approach was consistently applied across various thresholds, ensuring instances were labeled as "punch" only if they met the corresponding percentage criteria within the specified window duration. The 60% criteria outperformed other thresholds, resulting in higher accuracy. Using this method, the dataset was categorized into four classes: 'no punch,' 'long-range jab to the head (jab),' 'mid-range hook to the head (hook),' and 'mid-range uppercut to the head (uppercut),' for both rear and lead hand punches.

### 2.4.1. Active Learning Technique with Query Strategy: Query By Committee (QBC).

We have chosen the Query by Committee (QBC) based active learning technique due to its effectiveness in punch classification, particularly in scenarios where our previous study [6] using random forest or ensemble learning techniques showed that coaches or field experts face difficulty annotating 80% of the data to train the model. QBC is a robust active learning method that harnesses the combined intelligence of multiple weak learners, such as the Naive Bayes classifier, k-nearest neighbor, and decision tree. Compared to other strategies like uncertainty sampling and disagreement-based sampling, QBC offers key advantages: it ensures broader exploration and improves generalization by selecting samples based on the disagreement between models, making it ideal for the diverse punching styles in boxing. Additionally, it efficiently uses limited labeled data, is robust to noisy data, and adapts well as more data is labeled, allowing for continuous model improvement [24].

Steps Followed in Active Learning Using Query by Committee Technique:

1. Randomly select 5% of the dataset for initial training, reserving 95% for testing.

2. Utilize a weak learner committee (Naive Bayes classifier, k-nearest neighbor, decision tree, or ensemble learner) to train the model using the Query by Committee (QBC) strategy.

- The Bayes classifier, grounded in Bayesian probability theory, excels in the probabilistic classification of the punch.

$$P(C_k/X) = \frac{P\left(\frac{X}{C_k}\right).P(C_k)}{P(X)}$$

(1)

$P(C_k/X)$ = Posterior probability of class $C_k$ given spectro-temporal features X.

$P\left(\frac{X}{C_k}\right) =$ Likelihood of spectro-temporal features X given classes $C_k$.

$P(C_k)$ = Prior probability of punch class $C_k$.

$P(X)$ = Marginal probability of spectro-temporal features $X$.

- Decision trees, in the context of punch classification and recognition tasks, two common metrics used for splitting nodes in a decision tree are entropy and Gini impurity. Entropy is a measure of impurity or disorder in a set. Gini impurity quantifies the probability of incorrectly classifying an instance randomly chosen from the set. We used the default hyper-parameter settings, such as a Maximum number of splits being 100, and the split criterion being Gini's diversity index.

$$Entropy(S) = -\sum_{i=1}^{c} P_i.log_2(P_i) \tag{2}$$

$$Gini(S) = 1 - \sum_{i=1}^{c} P_i^2 \tag{3}$$

Where,

$S$ is the set of punches at a node.

$c$ is the number of classes (2 class for punch recognition and 3 class for punch classification)

$P_i$ is the probability of punch classWhere,

- KNN is particularly adept at capturing the punch signal local patterns and adapting to the underlying data distribution. Mathematically, if $X_{new}$ has spatiotemporal features of the punch $x_1, x_2, ....., x_n$, and $X_i$ is a data point in the training set with features $x_{i1}, x_{i2}, ....., x_{in}$, then the distance d between $X_{new}$ and $X_i$ can be calculated using Euclidean distance:

$$d(X_{new}, X_i) = \sqrt{\sum_{j=1}^{n} (x_{ij} - x_j)} \tag{4}$$

Where $x_j$ is the corresponding feature of $X_{new}$.

3. Average the output of the committee to classify predicted punches.

4. Compute entropy values for each sample using the Entropy sampling method.

- Entropy, a measure of uncertainty in information theory, is calculated for each sample of IoT sensor data, reflecting the diversity of punch class probabilities. High entropy values indicate instances of significant uncertainty, making them prime candidates for additional labeling.

$$\left(\frac{x}{M}\right) = \sum_c P_M^x(c).log\left(\frac{1}{P_M^x(c)}\right) \tag{5}$$

Where,

$M$ is the model

$P_M^x$ is the probability that sample $x$ belongs to punch class $c$ according to the model $M$.

5. Arrange entropy values in descending order alongside corresponding samples.

6. Identify samples with high entropy values, indicating uncertainty in classification.

7. Involve a domain expert or boxing coach to label uncertain samples for the next 5% of the dataset.

8. Add annotated data with the initial 5% training data. In Fig 5, the feature shown represents the model's uncertainty, determined by entropy. The Query-by-Committee (QBC) model assigns equal probabilities (0.5) to both jab and hook, indicating confusion or uncertainty. Consequently, the model queries the expert by asking, "What is the label of this instance?" Upon viewing the corresponding video frame for ground truth, the expert labels it as 1 for jab, 2 for hook, and 3 for uppercut. In this instance, the uncertain instance is identified as a jab, and the expert labels it as 1. This labeled data is then added to the training set, allowing the model to learn from it and avoid misclassification of similar features in the future.

9. Repeat the iterative process from step 2 to step 8 until the total training dataset reaches 15% to improve model accuracy and reduce uncertainty in predictions shown in Table 2.

Following model training, the system was applied to new athlete punch data to recognize punches and extract punch count and duration (start and end times). Initial attempts to determine punch duration by analyzing label transitions (0s for punches and 1s for no punches) faced challenges due to misclassifications, where 0s and 1s alternated within a punch event. To address this, the average punch duration of 0.8 seconds and a minimum interval of 0.2 seconds between events were used as constraints. Misclassified events within this interval were reverted to their previous state, ensuring accurate punch count and duration.

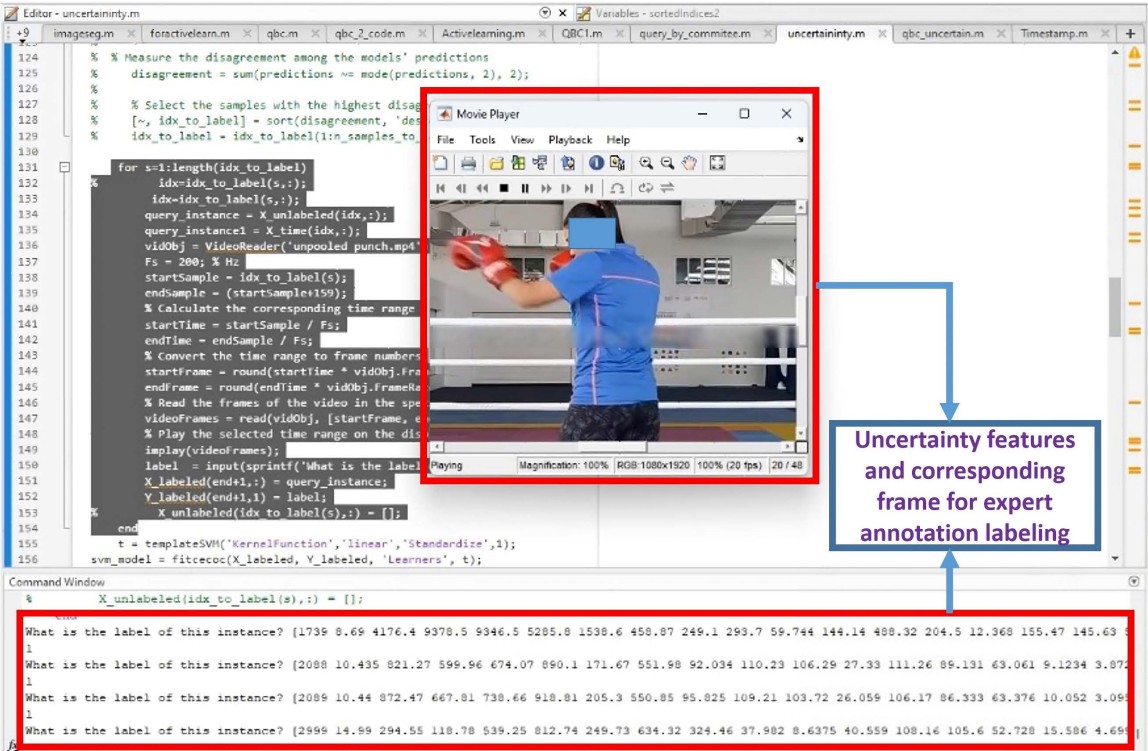

**Fig 5. Expert annotation of uncertain instances to refine model training.**

For hierarchical classification of punch types (jab, hook, uppercut) for rear and lead hands, the active learning technique was applied to train the model. The system was tested with new athlete data, where the mode value of output labels was used to assign a single label to each punch. This approach enabled accurate punch type classification and determination of punch counts for each category, with results summarized in Table 2.

## 3. Results

### 3.1. Automatic Punch Recognition

In the hierarchical classification framework, punch instances were identified within the dataset, with labels of 1 for punches and 0 for no punches, along with the corresponding start and end times. Confusion matrices in Fig 6 highlight misclassifications, primarily occurring during the initiation and conclusion phases of punches due to similarities between the preparatory phase (no punch) and these segments.

Table 2. Accuracy metrics with different percentage of the training dataset.

| Punch analytics | % of the training dataset | Accuracy | Precision | Recall | F1-Score |
|---|---|---|---|---|---|
| Punch Recognition | 5% | 83.17% | 83.67% | 84.92% | 0.84 |
| | 10% | 93.81% | 93.95% | 93.83% | 0.93 |
| | 15% | 94.19% | 95.02% | 94.23% | 0.95 |
| Punch Classification | 5% | 79.67% | 81.17% | 85.08% | 0.83 |
| | 10% | 90.86% | 91.44% | 91.80% | 0.91 |
| | 15% | 93.24% | 94.29% | 93.62% | 0.94 |

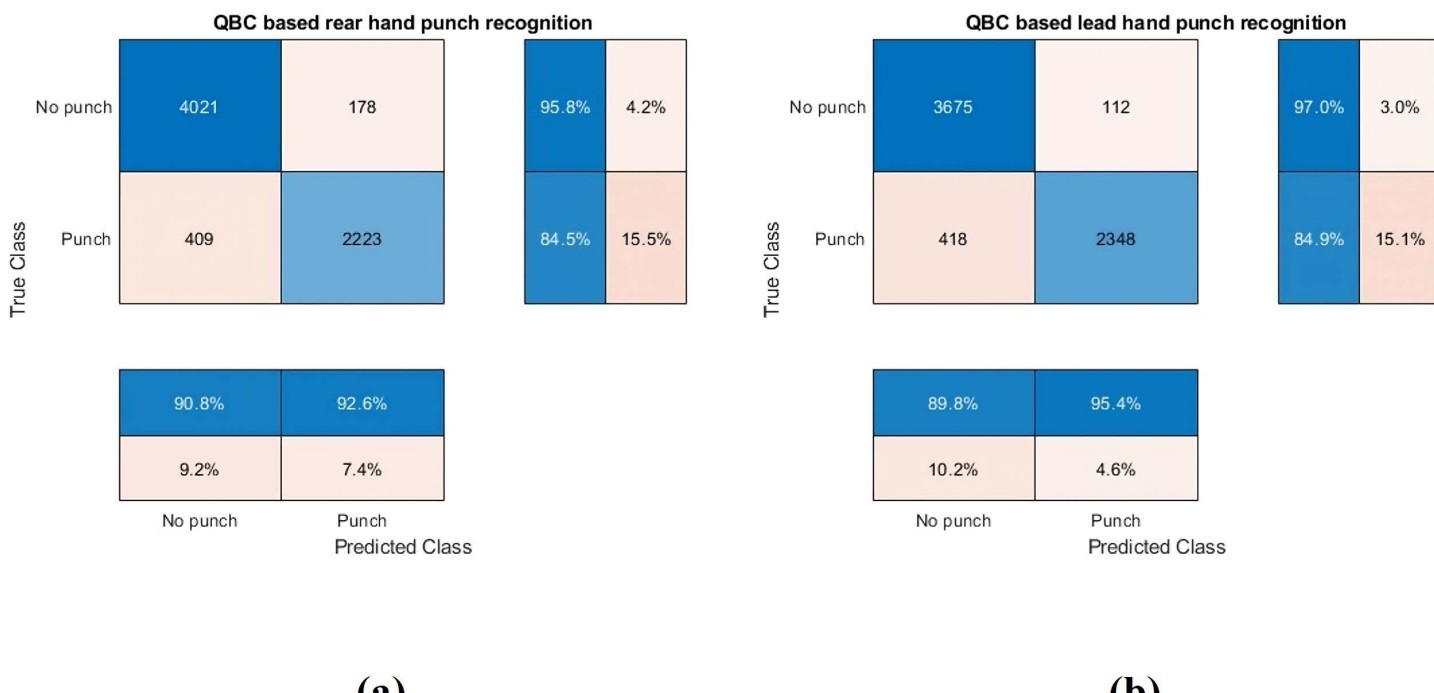

**(a)**                    **(b)**

**Fig 6. Combined Confusion matrix:** (a) Rear hand punch recognition (b) Lead hand punch recognition.

Accuracy metrics in Table 3 indicate punch recognition accuracies of 91.77% for the lead hand and 91.93% for the rear hand. These results were achieved using only 15% of the dataset, leveraging an active learning approach to optimize classification performance.

## 3.2. Punch classification

In the second stage of hierarchical classification, the system categorizes specific punch types—Jab, Hook, and Uppercut— for both lead and rear hands, building upon initial punch recognition. This classification uses an active learning technique with only 15% of the training dataset. Confusion matrices in Fig 7 demonstrate the model's high accuracy in classifying Jabs and Uppercuts, although some Hooks are misclassified as Jabs. Accuracy metrics are detailed in Table 3. A comparative statistical analysis of the misclassified Jab punches, revealed a closer resemblance to the true Jab category than the Hook category. This misclassification is likely due to similarities in the preparatory motion preceding punch execution.

## 3.3. Total punch count

A total of 17 punches have been identified in the provided dataset. However, the actual ground truth punch count is 18 as shown in the Fig 8. The discrepancy may be attributed to the occurrence of combined punches or rapidly repeated punches with a gap of less than or equal to 0.2 seconds.

## 3.4. Punch start and end time

Following the classification process, the commencement and conclusion times of punches are derived from the predicted class labels, which are represented as a series of 0's (No punch) and 1's (Punch). The initiation time of a punch is determined by identifying the start of a consecutive series of 1's, while the conclusion time corresponds to the termination of that series. The total punch count is obtained by counting the unique sets of start and end times. The punch times, as shown in Fig 9(a)–(c), are highlighted in yellow above the boxer's head.

These times are critical for comparing elite boxers, enabling the identification of the most effective and fastest punches. By analyzing these punches, we can train athletes to optimize their punch combinations, ultimately improving their performance and increasing their chances of defeating opponents.

**Table 3. Model testing results across different boxers with 15% of training dataset.**

| Punch analytics | Subject | Hand | Accuracy | Precision | Recall | F1-Score |
|---|---|---|---|---|---|---|
| Punch recognition | 1 | Rear | 91.34% | 91.34% | 91.27% | 0.91 |
| | | Lead | 92.31% | 92.83% | 91.63% | 0.92 |
| | 2 | Rear | 92.58% | 91.97% | 91.06% | 0.92 |
| | | Lead | 91.58% | 92.35% | 90.40% | 0.91 |
| | Mixed | Rear | 91.41% | 91.68% | 90.11% | 0.91 |
| | | Lead | 91.91% | 92.62% | 90.97% | 0.92 |
| Punch classification | 1 | Rear | 97.46% | 97.75% | 97.51% | 0.98 |
| | | Lead | 93.87% | 94.60% | 94.16% | 0.94 |
| | 2 | Rear | 90.59% | 92.03% | 90.64% | 0.91 |
| | | Lead | 89.94% | 91.35% | 91.08% | 0.90 |
| | Mixed | Rear | 92.33% | 93.25% | 92.58% | 0.93 |
| | | Lead | 94.56% | 94.68% | 95.14% | 0.95 |

### 3.5. Punch type count

Here, we aim to determine the types of punches and their respective counts in a bout. This is achieved by utilizing punches' start and end times, where the predominant occurrences in the predicted labels during a specific duration indicate the types of punches. The confusion matrix, as illustrated in Fig 8, serves as a valuable tool for discerning the count of each punch type in the classification process, and its accuracy is about 94%.

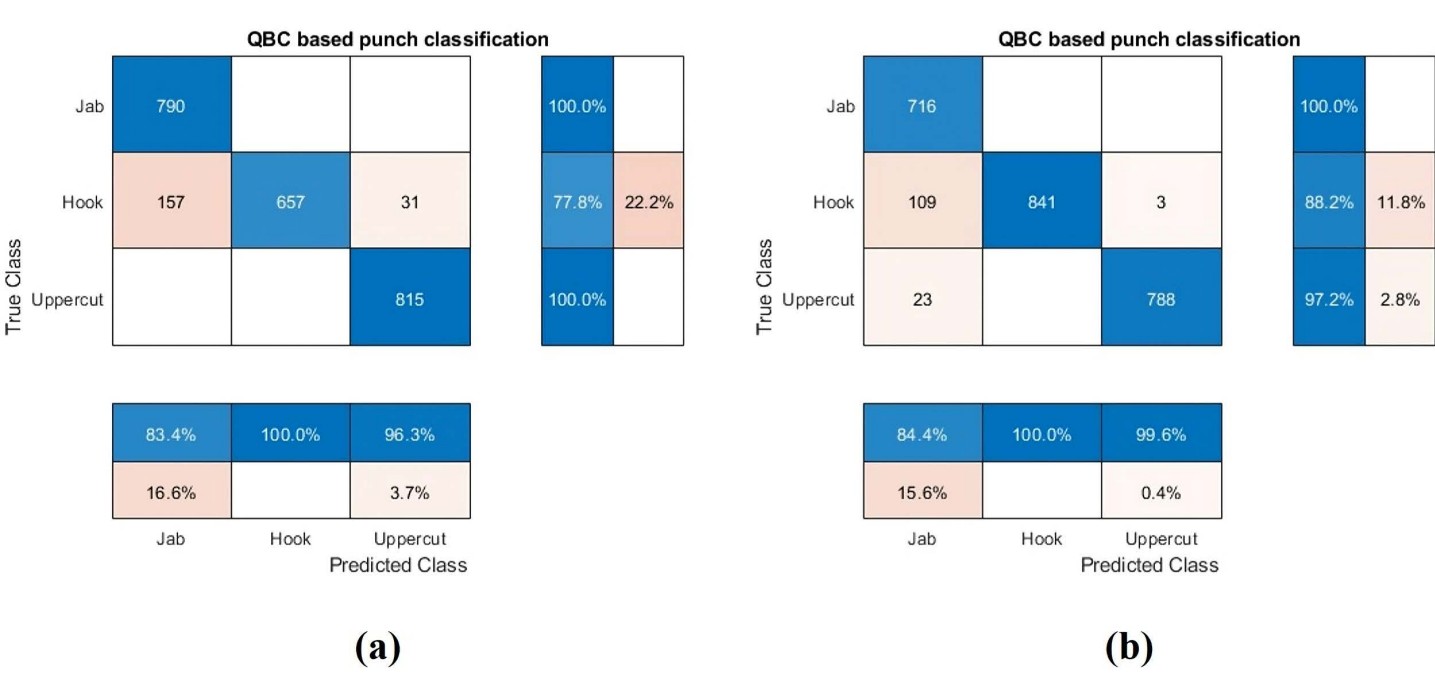

(a) (b)

**Fig 7. Combined Confusion matrix: (a) Rear hand punch classification (b) Lead hand punch classification.**

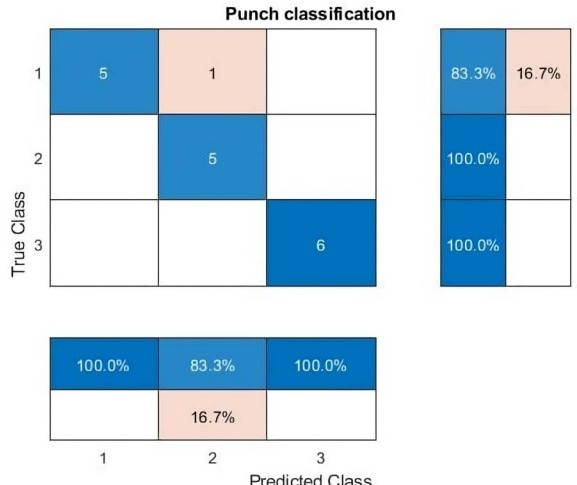

**Fig 8. Confusion matrix-punch classification.**

### 3.6. Punch frame segmentation

We segmented the punch frames in the video using the start and end times of the punches. Fig 9 illustrates the punch segmentation for all punches. In this representation, we have selected frames, including the starting time frame, the frame where the punch occurs, and the ending time frame, rather than showcasing every single frame.

## 4. Discussion

Our study employs a Query-by-Committee (QBC) active learning technique utilizing a combination of Bayes classifier, decision tree, and k-nearest neighbors (KNN), enabling the model to be trained with only 15% of the available training data. High accuracy is achieved by training uncertain data points, identified via entropy-based uncertainty sample selection. This method contrasts with existing literature, which predominantly employs machine learning and deep learning algorithms such as Support Vector Classifier (SVC), Multi-Layer Perceptron (MLP), Dynamic Time Warping (DTW), Convolutional Neural Networks (CNNs), and Random Forest, typically utilizing 70–80% of the training data to achieve similar levels of accuracy. Automatic punch detection and classification were successfully accomplished using this active learning technique, validated with data from an unknown boxer, highlighting its efficiency and innovative potential. Traditional approaches, as shown in Table 4, require of training data, whereas our study achieved remarkable results using only a minimal dataset. our approach reduces the need for extensive data acquisition and suggests real-world applicability, particularly in challenging scenarios.

Our model demonstrates significant strengths by achieving competitive punch recognition and classification accuracy by using only 36 punches (both lead and rear hand) per type for training, the model performs effectively even in situations

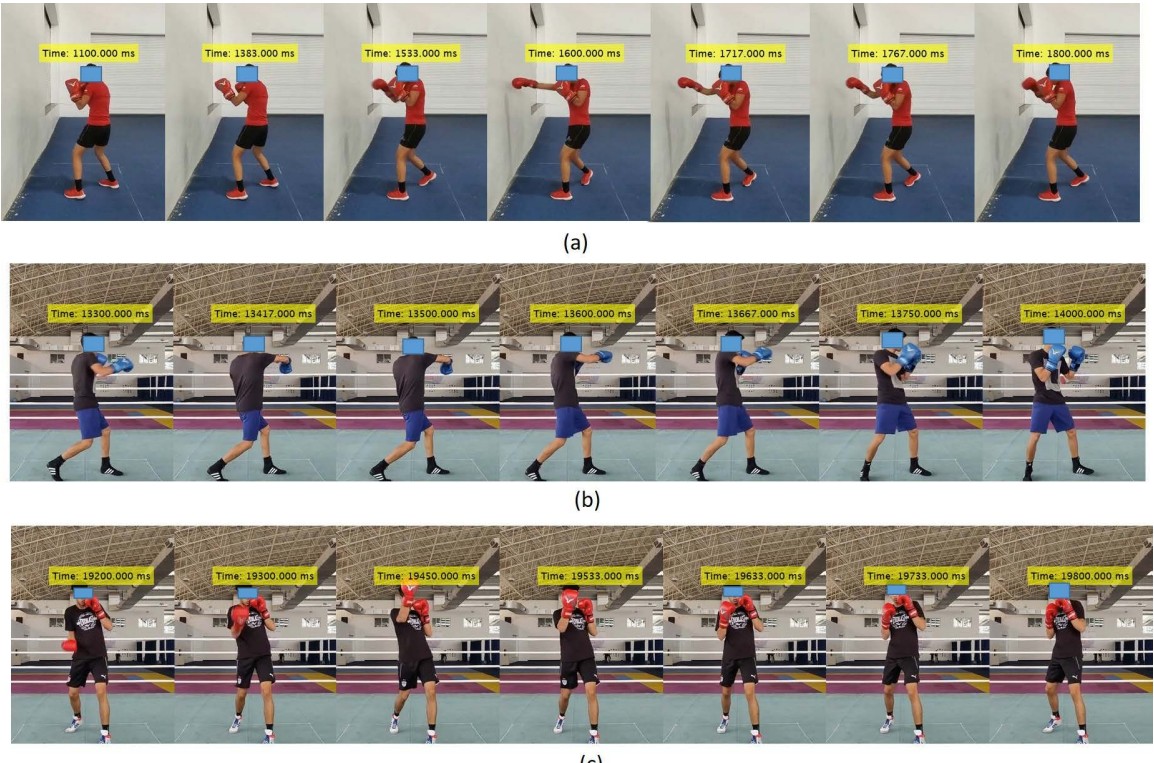

**Fig 9. Validation of punch with timestamp: (a) Jab (b) Hook (c) Uppercut.**

**Table 4. Literature comparison.**

| Reference | Punch count | % of data for model training | Algorithm used | Punch recognition | Punch classification | Punch Duration | Punch video segmentation |
|---|---|---|---|---|---|---|---|
| [25] | Jab-776, Hook-434, Uppercut-475, Backfist- 139 | 70-80% | Support vector classifier (SVC) | – | 93.03% (striking technique) | ✓ | – |
| [13] | Lead and rear jab-6000, Lead and Rear hook-6000, Lead and uppercut- 6000 | 70-80% | Multilayer Perceptron (MLP) | 87% to 99% | 81% to 97% | – | – |
| [8] | Lead and rear jab-100, Lead and rear hook-100, Lead and uppercut-100 | 70-80% | Multilayer Perceptron (MLP) | – | 98% | – | – |
| [26] | Jab, hook, uppercut- 2880 | 70-80% | Dynamic Time warping (DTW) and CNNs | – | 92.08% | – | – |
| [6] | Jab-120, hook-120, uppercut- 120 | 80% | Random forest | – | 96% | – | – |
| **Our model** | **Lead and rear jab-36, Lead and Rear hook-36, Lead and uppercut- 36** | **15%** | **Query by committee based active learning technique** | **91.41% for rear hand and 91.91% for lead hand** | **92.33% for rear hand and 94.56% lead hand** | ✓ | ✓ |

with limited data availability. It maintains a strong balance in accuracy for both lead and rear hand punches, showing its reliability in recognizing different striking techniques. Additionally, the implementation of the Query By Committee (QBC) based active learning technique enhances the model's ability to focus on the most informative data points, improving performance while minimizing the need for extensive labeled data. This efficiency and adaptability make our model a suitable solution for punch classification and recognition in training and performance monitoring settings.

An inherent limitation arises from the challenge of accurately predicting the start and end times of punch segmentation, primarily because the initiation and conclusion portions of all punches share a similar style, with only the intermediate portion providing distinction among different punch types. Another limitation lies in the system's inability to independently identify combinations of punches occurring within a time gap of less than or equal to 0.2 seconds, as in Fig 9(a). The IoT wearable sensor can accurately segment the punches during practice matches without any restrictions, regardless of the boxer's direction of movement also automatic segment the each punch video clips with timestamp. This technology eliminates the disadvantages associated with computer vision systems, such as blind spots or occlusions, as shown in Fig 9(b).

## 5. Conclusions

This study utilizes active learning techniques for automatic punch recognition and classification using IMU sensor data, providing key metrics such as total punch count, start and end times, and the count of specific punch types. The proposed method integrates IMU and video data, automatically extracting video segments corresponding to punches, thus enhancing its utility in training environments. These insights assist coaches and boxers in identifying strengths, weaknesses, and areas for technique and strategy refinement, while also reducing the labeling effort required from domain experts. The system has been demonstrated to a boxing coach, who utilized it to correct the boxer by reviewing the video and addressing specific mistakes. Boxers have shown interest in the video-based analytics, finding it beneficial for improving their performance and easily understanding their areas for improvement. Additionally, while the study primarily focuses on elite boxers, the proposed method is applicable to amateur and grassroots athletes. Since the model is trained on amateur

boxer data, it inherently captures a range of punching styles while maintaining key biomechanical patterns. As long as the fundamental punching mechanics are followed—such as forward and backward motion for jabs, left and right angular movement for hooks, and upward trajectory for uppercuts—the model can accurately recognize and classify punches. However, extreme variations in technique, such as unstructured or incorrect movements, may impact classification accuracy. Future work will focus on further developing the Smart Boxer system by integrating IMU sensors and computer vision for real-time bout analysis. The system will be expanded to classify all 14 punch types and incorporate additional metrics such as punch force and kinetic chain analysis. This approach aims to provide comprehensive performance insights, optimize training strategies, and reduce injury risks, representing a significant advancement in boxing performance analysis.

6. Acknowledgments I am grateful to our co-authors for their continuous guidance and support, as well as to Inspire Institute of Sports (IIS) academy, Centre of Excellence for Sports Science and Analytics and our Human Cyber Physical System (HCPS) labmates for their collaboration and contributions. Thank you all for being part of this journey and for your dedication to advancing knowledge in our field.

## Author contributions

**Conceptualization:** Saravanan Manoharan, Ravi Sadananda Hegde, Ranganathan Srinivasan, Babji Srinivasan.

**Data curation:** Saravanan Manoharan, John Warburton.

**Formal analysis:** John Warburton, Ravi Sadananda Hegde, Ranganathan Srinivasan, Babji Srinivasan.

**Funding acquisition:** Ravi Sadananda Hegde, Ranganathan Srinivasan, Babji Srinivasan.

**Investigation:** John Warburton, Ravi Sadananda Hegde, Ranganathan Srinivasan, Babji Srinivasan.

**Methodology:** Saravanan Manoharan, Babji Srinivasan.

**Resources:** Saravanan Manoharan, John Warburton.

**Software:** Saravanan Manoharan.

**Supervision:** Ravi Sadananda Hegde, Ranganathan Srinivasan, Babji Srinivasan.

**Validation:** Saravanan Manoharan, Ranganathan Srinivasan, Babji Srinivasan.

**Visualization:** Saravanan Manoharan.

**Writing – original draft:** Saravanan Manoharan.

**Writing – review & editing:** Ravi Sadananda Hegde, Ranganathan Srinivasan, Babji Srinivasan.

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
