## [Decision Letter · Decision Letter 0]

26 Feb 2025

PONE-D-25-04904An Active Machine Learning Framework for Automatic Boxing Punch Recognition and Classification Using Upper Limb KinematicsPLOS ONE

Dear Dr. Srinivasan,

Thank you for submitting your manuscript to PLOS ONE. After careful consideration, we feel that it has merit but does not fully meet PLOS ONE’s publication criteria as it currently stands. Therefore, we invite you to submit a revised version of the manuscript that addresses the points raised during the review process.

We look forward to receiving your revised manuscript.

Kind regards,

Xiaohui Zhang

Academic Editor

PLOS ONE

Journal Requirements:

This research was supported by the Centre of Excellence for Sports Science and Analytics for funding from the Indian Institute of Technology, Madras, under Grant SP22231231CPETWOSSAHOC.

6. We note that you have indicated that there are restrictions to data sharing for this study. For studies involving human research participant data or other sensitive data, we encourage authors to share de-identified or anonymized data. However, when data cannot be publicly shared for ethical reasons, we allow authors to make their data sets available upon request. For information on unacceptable data access restrictions, please see http://journals.plos.org/plosone/s/data-availability#loc-unacceptable-data-access-restrictions.

7. In the online submission form, you indicated that the datasets collected and analyzed during the current study are available from the corresponding author upon reasonable request.

Reviewers' comments:

Reviewer's Responses to Questions

**Comments to the Author**

1. Is the manuscript technically sound, and do the data support the conclusions?

Reviewer #1: Yes

Reviewer #2: Yes

2. Has the statistical analysis been performed appropriately and rigorously? 

Reviewer #1: Yes

Reviewer #2: Yes

3. Have the authors made all data underlying the findings in their manuscript fully available?

Reviewer #1: Yes

Reviewer #2: Yes

4. Is the manuscript presented in an intelligible fashion and written in standard English?

Reviewer #1: Yes

Reviewer #2: Yes

5. Review Comments to the Author

Reviewer #1: The paper presents an innovative approach combining wearable sensors and video data with active learning techniques for punch classification in boxing. The method is compelling, with a more detailed discussion on its comparative advantages over deep learning-based techniques would strengthen the manuscript.

(1) While the introduction provides a strong justification for the study, it would be beneficial to include a discussion on limitations of existing work and how this study addresses them more explicitly. In addition, it would be better to discuss how the proposed method compares with deep learning approaches in terms of performance, computational cost, and real-time applicability.

(2) The study mentions elite boxers but does not discuss the generalizability of the approach to amateur or lower-level athletes. Would the model generalize well in a different dataset?

(3) The manuscript describes extracted features, but it would help to clarify why the specific statistical features were chosen and how they contribute to classification.

(4) While the Query by Committee (QBC) method is well explained, it can be better to discuss about the rationale behind choosing this particular active learning strategy over alternatives (e.g., uncertainty sampling, disagreement-based sampling).

Reviewer #2: The paper presents an approach that employs multiple models for automatic punch recognition and classification. The mathematical analysis is thorough, and the experimental results demonstrate the effectiveness of the proposed method. However, the approach largely builds upon and integrates existing techniques for boxing punch recognition, a niche application. Consequently, the method lacks a high degree of innovation.

6. PLOS authors have the option to publish the peer review history of their article (what does this mean? ). If published, this will include your full peer review and any attached files.

**Do you want your identity to be public for this peer review?** For information about this choice, including consent withdrawal, please see our Privacy Policy .

Reviewer #1: No

Reviewer #2: No

---

## [Author Response · Author response to Decision Letter 0]

4 Mar 2025

An Active Machine Learning Framework for Automatic Boxing Punch Recognition and Classification Using Upper Limb Kinematics

Saravanan Manoharan, John Warburton, Ravi Sadananda Hegde, Ranganathan Srinivasan and Babji Srinivasan

Response to Reviewers’ Comments

The authors would like to thank the reviewers at the outset for their valuable comments which have helped us in improving our manuscript significantly. We have highlighted the overall changes made in manuscript in order to improve it.

Reply to Reviewers

Reviewer 1

1) Reviewer’s comment

While the introduction provides a strong justification for the study, it would be beneficial to include a discussion on limitations of existing work and how this study addresses them more explicitly. In addition, it would be better to discuss how the proposed method compares with deep learning approaches in terms of performance, computational cost, and real-time applicability.

Response:

• Unlike deep learning models, which require 70-80% labeled training data, our approach uses query-by-committee active learning, requiring only 15% of the dataset while maintaining high accuracy.

• Deep learning models like CNNs and MLPs often face overfitting issues with limited data, whereas our method improves adaptability by iteratively refining predictions.

• While deep learning models can be computationally expensive, our method is more efficient, making it feasible for post-processing punch analysis without requiring high-end hardware.

In view of this comment, highlighted portion of the following text is added in Section 1.2 of the revised manuscript.

A query-by-committee-based active learning technique minimizes labeled data requirements, using only 15% for training while improving accuracy and adaptability. Unlike deep learning models, which typically require 70-80% of the dataset for training, our approach leverages active learning to minimize data labeling efforts while maintaining high classification accuracy. Deep learning models, such as CNNs and MLPs, often face overfitting issues when training data is limited, whereas our method iteratively refines predictions, improving adaptability. Furthermore, deep learning approaches can be computationally expensive, making real-time processing and post-activity punch analysis challenging without high-end hardware. Our approach is more computationally efficient, ensuring practical implementation for both training and competition scenarios. Also identifies key performance indicators (KPIs), including total punch count, punch start and end times, and punch type categorization, offering detailed insights into a boxer’s performance and activity levels.

2) Reviewer’s comment

The study mentions elite boxers but does not discuss the generalizability of the approach to amateur or lower-level athletes. Would the model generalize well in a different dataset?

Response: The proposed method generalizes well to amateur and lower-level athletes since the model is trained on amateur boxer data, capturing a range of punching styles while maintaining fundamental biomechanical patterns. As long as punches follow the correct line of action—jab (forward/backward), hook (left/right angular), and uppercut (upward)—the model can accurately recognize and classify them. However, extreme variations in technique or unstructured movements may impact classification performance. Ensuring adherence to fundamental punching mechanics enhances accuracy across all skill levels.

In view of this comment, highlighted portion of the following text is added in Section 5 (conclusion) of the revised manuscript.

Boxers have shown interest in the video-based analytics, finding it beneficial for improving their performance and easily understanding their areas for improvement. Additionally, while the study primarily focuses on elite boxers, the proposed method is applicable to amateur and grassroots athletes. Since the model is trained on amateur boxer data, it inherently captures a range of punching styles while maintaining key biomechanical patterns. As long as the fundamental punching mechanics are followed—such as forward and backward motion for jabs, left and right angular movement for hooks, and upward trajectory for uppercuts—the model can accurately recognize and classify punches. However, extreme variations in technique, such as unstructured or incorrect movements, may impact classification accuracy. Future work will focus on further developing the Smart Boxer system by integrating IMU sensors and computer vision for real-time bout analysis.

3. Reviewer’s comment

The manuscript describes extracted features, but it would help to clarify why the specific statistical features were chosen and how they contribute to classification.

Response:

The selected statistical and time-frequency features were chosen based on the distinct kinematic characteristics of each punch type, ensuring accurate classification:

• Time-domain features (mean, standard deviation, max, min, interquartile range, entropy, skewness, kurtosis, mean absolute deviation) capture variations in punch intensity, curvature, and rotational control.

• Frequency-domain features (power spectral density, spectrogram) identify dominant motion patterns across axes, with jabs showing higher spectral density on the x-axis, hooks on the y-axis, and uppercuts on the z-axis.

In view of this comment, highlighted portion of the following text is added in Section 2.3 (Feature extraction) of the revised manuscript.

The selected statistical and time-frequency features were chosen based on the distinct kinematic characteristics of each punch type, ensuring accurate classification: Time-domain features (mean, standard deviation, max, min, interquartile range, entropy, skewness, kurtosis, mean absolute deviation) capture variations in punch intensity, curvature, and rotational control. Frequency-domain features (power spectral density, spectrogram) identify dominant motion patterns across axes, with jabs showing higher spectral density on the x-axis, hooks on the y-axis, and uppercuts on the z-axis.

4. Reviewer’s comment

While the Query by Committee (QBC) method is well explained, it can be better to discuss about the rationale behind choosing this particular active learning strategy over alternatives (e.g., uncertainty sampling, disagreement-based sampling).

Response: We chose Query by Committee (QBC) over other active learning strategies like uncertainty sampling and disagreement-based sampling due to its advantages in our study. QBC utilizes multiple models to identify disagreement, ensuring broader exploration and improved generalization, which is crucial for boxing punch classification given the variation in athlete techniques. It efficiently selects the most informative data points, enhancing model performance with only 15% labeled data, unlike uncertainty sampling, which may focus too narrowly on uncertain examples. Additionally, QBC helps improve robustness to noisy and diverse boxing data by exposing the model to uncertain areas, while disagreement-based sampling focuses solely on individual model disagreements. Moreover, QBC scales well with increasing labeled data, continuously improving model performance by selecting diverse and informative samples for labeling. This strategy maximizes the effectiveness of the model, especially when working with challenging sports data.

In view of this comment, highlighted portion of the following text is added in Section 2.4.1 (Active Learning Technique with Query Strategy: Query By Committee (QBC)) of the revised manuscript.

QBC is a robust active learning method that harnesses the combined intelligence of multiple weak learners, such as the Naive Bayes classifier, k-nearest neighbor, and decision tree. Compared to other strategies like uncertainty sampling and disagreement-based sampling, QBC offers key advantages: it ensures broader exploration and improves generalization by selecting samples based on the disagreement between models, making it ideal for the diverse punching styles in boxing. Additionally, it efficiently uses limited labeled data, is robust to noisy data, and adapts well as more data is labeled, allowing for continuous model improvement [24].

Reviewer 2

Reviewer’s comment

The paper presents an approach that employs multiple models for automatic punch recognition and classification. The mathematical analysis is thorough, and the experimental results demonstrate the effectiveness of the proposed method. However, the approach largely builds upon and integrates existing techniques for boxing punch recognition, a niche application. Consequently, the method lacks a high degree of innovation

Response: Our approach incorporates existing techniques for punch recognition but introduces several innovations that improve its functionality and applicability. First, the integration of active learning through the Query-by-Committee (QBC) method enables the system to learn efficiently from only 15% of labeled data, significantly reducing the annotation effort required from domain experts. The use of entropy-based uncertainty sampling within QBC further enhances the system's learning efficiency by focusing on the most informative data points, which is crucial for optimizing the model’s performance in the fast-paced nature of boxing. This innovation makes the system more efficient and applicable in environments with limited labeled data. Additionally, the system achieves high accuracy with a small training dataset (36 punches per type), whereas traditional methods require larger labeled datasets. Furthermore, the multimodal integration of IMU sensor data and video clips addresses challenges in both video-based analysis (such as motion blur, reduced classification accuracy, multi-camera synchronization, and blind spot issues) and sensor limitations (such as insufficient visual feedback and challenges in learning or training directly from video). This combination provides a more comprehensive analysis tool for coaches and boxers, allowing for automatic punch recognition, classification, and segmentation of bout videos. The method has been validated on data from unknown boxers, and positive feedback from coaches and boxers has been received, confirming its practical use. These contributions provide a more efficient, scalable, and advanced approach to boxing punch analysis compared to previous methods.

In view of this comment, highlighted portion of the following text is added in Section 1.2 of the revised manuscript.

Vision systems provide real-time bout analysis but have motion blur, reduced classification accuracy, multi-camera synchronization, blind spot issues, and lower accuracy, potentially affecting punch classification

---

## [Decision Letter · Decision Letter 1]

23 Mar 2025

An Active Machine Learning Framework for Automatic Boxing Punch Recognition and Classification Using Upper Limb Kinematics

PONE-D-25-04904R1

Dear Dr. Srinivasan,

We’re pleased to inform you that your manuscript has been judged scientifically suitable for publication and will be formally accepted for publication once it meets all outstanding technical requirements.

Kind regards,

Xiaohui Zhang

Academic Editor

PLOS ONE

Additional Editor Comments (optional):

Reviewers' comments:

Reviewer's Responses to Questions

**Comments to the Author**

1. If the authors have adequately addressed your comments raised in a previous round of review and you feel that this manuscript is now acceptable for publication, you may indicate that here to bypass the “Comments to the Author” section, enter your conflict of interest statement in the “Confidential to Editor” section, and submit your "Accept" recommendation.

Reviewer #1: All comments have been addressed

Reviewer #2: All comments have been addressed

2. Is the manuscript technically sound, and do the data support the conclusions?

Reviewer #1: Yes

Reviewer #2: Yes

3. Has the statistical analysis been performed appropriately and rigorously? 

Reviewer #1: Yes

Reviewer #2: Yes

4. Have the authors made all data underlying the findings in their manuscript fully available?

Reviewer #1: Yes

Reviewer #2: Yes

5. Is the manuscript presented in an intelligible fashion and written in standard English?

Reviewer #1: Yes

Reviewer #2: Yes

6. Review Comments to the Author

Reviewer #1: The authors have addressed all the comments from the reviewers. The paper is good for publication now.

Reviewer #2: In the revised version, the authors highlighted the creativity and potential applications of the method in other areas. They also provided a rationale for their choice to employ this method. The paper meets the standards required for publication in PLOS ONE.

7. PLOS authors have the option to publish the peer review history of their article (what does this mean? ). If published, this will include your full peer review and any attached files.

**Do you want your identity to be public for this peer review?** For information about this choice, including consent withdrawal, please see our Privacy Policy .

Reviewer #1: No

Reviewer #2: No

---

## [Editor Report · Acceptance letter]

PONE-D-25-04904R1

PLOS ONE

Dear Dr. Srinivasan,

I'm pleased to inform you that your manuscript has been deemed suitable for publication in PLOS ONE. Congratulations! Your manuscript is now being handed over to our production team.

Kind regards,

on behalf of

Dr. Xiaohui Zhang

Academic Editor

PLOS ONE